# Ancient co-option of an amino acid ABC transporter locus in *Pseudomonas syringae* for host signal-dependent virulence gene regulation

**Qing Yan**[1¤], **Conner J. Rogan**[1], **Yin-Yuin Pang**[1], **Edward W. Davis, II**[2], **Jeffrey C. Anderson**[1]*

**1** Department of Botany and Plant Pathology, Oregon State University, Corvallis, Oregon, United States of America, **2** Center for Genome Research and Biocomputing, Oregon State University, Corvallis, Oregon, United States of America

¤ Current address: Department of Plant Sciences and Plant Pathology, Montana State University, Bozeman, Montana, United States of America
* anderje2@oregonstate.edu

**Data Availability Statement:** All relevant data are within the manuscript and Supporting Information data file.

## Abstract

Pathogenic bacteria frequently acquire virulence traits via horizontal gene transfer, yet additional evolutionary innovations may be necessary to integrate newly acquired genes into existing regulatory pathways. The plant bacterial pathogen *Pseudomonas syringae* relies on a horizontally acquired type III secretion system (T3SS) to cause disease. T3SS-encoding genes are induced by plant-derived metabolites, yet how this regulation occurs, and how it evolved, is poorly understood. Here we report that the two-component system AauS-AauR and substrate-binding protein AatJ, proteins encoded by an *acidic amino acid-transport* (*aat*) and *-utilization* (*aau*) locus in *P. syringae*, directly regulate T3SS-encoding genes in response to host aspartate and glutamate signals. Mutants of *P. syringae* strain DC3000 lacking *aauS*, *aauR* or *aatJ* expressed lower levels of T3SS genes in response to aspartate and glutamate, and had decreased T3SS deployment and virulence during infection of Arabidopsis. We identified an AauR-binding motif (Rbm) upstream of genes encoding T3SS regulators HrpR and HrpS, and demonstrated that this Rbm is required for maximal T3SS deployment and virulence of DC3000. The Rbm upstream of *hrpRS* is conserved in all *P. syringae* strains with a canonical T3SS, suggesting AauR regulation of *hrpRS* is ancient. Consistent with a model of conserved function, an *aauR* deletion mutant of *P. syringae* strain B728a, a bean pathogen, had decreased T3SS expression and growth in host plants. Together, our data suggest that, upon acquisition of T3SS-encoding genes, a strain ancestral to *P. syringae* co-opted an existing AatJ-AauS-AauR pathway to regulate T3SS deployment in response to specific host metabolite signals.

**Funding:** This study was funded by National Science Foundation award IOS-1557694 to JCA. The funders had no role in study design, data collection and analysis, decision to publish, or preparation of the manuscript.

**Competing interests:** The authors have declared that no competing interests exist.

## Author summary

Compared to well-established models of plant immunity, little is known about how plant pathogenic microorganisms detect their hosts to deploy virulence factors. Here we identify a two-component system AauS-AauR in the plant pathogenic bacterium *Pseudomonas syringae* that links perception of specific plant-exuded amino acids directly to regulation of genes encoding a virulence-promoting type III secretion system. AauS-AauR also functions in amino acid transport and is present in non-pathogenic pseudomonads, indicating that *P. syringae* has repurposed this pathway for virulence. Comparative genomics suggests that all pathogenic *P. syringae* rely on AauS-AauR signaling for T3SS deployment during infection. Therefore, future efforts to inhibit this pathway may be an effective strategy to curb diseases caused by *P. syringae* across its broad host range.

## Introduction

Outcomes of host-pathogen interactions are often determined by complex reciprocal signaling events that occur at the start of infection. Typically, a successful pathogen must recognize the presence of a host to begin deploying virulence factors, whereas to prevent infection, the host must recognize the invader and respond with effective defenses. Because defense and virulence factors are energetically costly to produce and can decrease fitness in the absence of interaction [1], many hosts and pathogens keep their respective defense and virulence programs in check prior to interaction, yet capable of rapid deployment. In this regard, plants have evolved receptors that constantly surveil the extracellular space for pathogen/microbe-associated molecular patterns (PAMPs/MAMPs) [2], and these receptors can trigger effective defenses when activated [3]. Many PAMPs and their respective plant receptors are known, and molecular details of PAMP-induced signaling are extensively studied [3]. In contrast, very little is known about how plant pathogens detect host signals to upregulate genes required for pathogenesis [4].

*Pseudomonas syringae* are Gram-negative bacteria that, as a species complex, can infect many plants including economically-important crops, and serve as model pathogens for studies of the molecular basis of pathogenesis and disease resistance [5]. A key virulence determinant of *P. syringae* is the type III secretion system (T3SS), a syringe-like apparatus that delivers effector proteins into host cells to suppress PAMP-activated defense signaling and other defense-associated cellular responses [6–8]. The T3SS must be produced *de novo* at the start of infection. A metabolomics analysis identified specific plant-exuded metabolites, including aspartate, citrate and 4-hydroxybenzoate, as potent inducers of T3SS-encoding genes in the model strain *P. syringae* pv. *tomato* DC3000 [9]. The importance of these metabolite signals for infection outcomes was demonstrated by Arabidopsis *mapk phosphatase 1* (*mkp1*), a mutant that exudes lower amounts of these metabolites and consequently is more resistant to infection [9].

Proteins that regulate T3SS deployment by *P. syringae* include the alternative sigma factor HrpL that binds to conserved *hrp* box sequences within the promoters of T3SS-encoding genes to upregulate their expression [10]. In turn, the expression of *hrpL* is regulated by two NtrC-like enhancer binding proteins HrpR and HrpS that bind the *hrpL* promoter and recruit the sigma factor RpoN ($\sigma^{54}$) [10–14]. It is not clear how the HrpR/HrpS-HrpL node, central to regulating type III secretion, is activated by host signals. HrpR and HrpS are expressed from a single operon that can be expressed under both T3SS-inducing and non-inducing conditions [10, 12, 13, 15]. Further, both HrpR and HrpS lack signal receiver domains commonly found in NtrC-like proteins, suggesting they are not directly regulated by intracellular signals [10,

11]. Proteins that regulate HrpR and HrpS transcriptionally and post-transcriptionally are known [15–17], yet are unlikely to be directly involved in perceiving T3SS-inducing host signals.

In this work we identified an ABC transporter-associated two-component system Aau-S-AauR and substrate-binding protein AatJ in *P. syringae* that directly connect perception of host-derived acidic amino acids to transcriptional regulation of T3SS-encoding genes. We show that AauR regulates *hrpRS* expression through binding to a conserved cis regulatory motif within the *hrpRS* promoter, and that this motif is conserved in essentially all pathogenic *P. syringae*. Because AauS-AauR and AatJ are present in all pseudomonads including non-pathogens, our data suggest that the transporter-associated functions of AatJ-AauS-AauR were co-opted for T3SS regulation early in *P. syringae* evolution and that this host signal-responsive signaling pathway has been maintained throughout subsequent evolution and host diversification of *P. syringae*.

## Results

### AauS-AauR and AatJ are required for maximal expression of T3SS-associated genes in DC3000

Multiple plant-exuded organic acids and amino acids are, in the presence of sugars, inducers of T3SS-encoding genes in *P. syringae* strain DC3000 [9]. To investigate the molecular basis of how these metabolites are perceived by *P. syringae*, we previously screened a population of 20,000 Tn*5*-mutagenized DC3000 colonies carrying an *avrRpm1*<sub>promoter</sub>:*gfp* reporter for decreased GFP fluorescence in response to fructose and aspartate [18]. Among mutants isolated by this screen, we identified four Tn*5* insertions within the *amino acid transport/ amino acid uptake* (*aat/aau*) locus encoding a predicted ABC transporter (AatQMP) and two-component system (AauSR) (**Fig 1A**). Three of the Tn*5* insertions were located within genes encoding the predicted sensor kinase (*aauS*) and response regulator (*aauR*), whereas the fourth Tn*5* insertion was located within *aatJ*, a predicted periplasmic substrate-binding protein (SBP). To verify the role of these genes in *avrRpm1* expression, we generated three deletion mutants that individually lack *aauS*, *aauR*, and *aatJ*, and tested these mutants for response to T3SS-inducing metabolites. Fructose alone is sufficient to induce T3SS gene expression [18–19], whereas aspartate acts synergistically with fructose to induce hyper-expression of T3SS-associated genes [9]. In response to fructose only, relative to each other and DC3000, each of the three deletion mutants had similar levels of expression of T3SS effector *avrRpm1* based on GFP fluorescence from an *avrRpm1*<sub>promoter</sub>:*gfp* reporter plasmid (**Fig 1B**). In contrast, all three deletion strains showed a clear reduction in *avrRpm1* expression when aspartate was also included (**Fig 1B**). We also generated a DC3000 mutant with both *aauS* and *aauR* deleted. Consistent with a model of AauS and AauR acting in the same signaling pathway, the level of *avrRpm1* expression in the Δ*aauS*Δ*aauR* double mutant was similar to *avrRpm1* expression levels in Δ*aauS* and Δ*aauR* strains (**Fig 1B**).

To investigate if *aauS*, *aauR* and *aatJ* regulate T3SS signaling pathways upstream of *avrRpm1*, we measured the expression of genes for T3SS regulators HrpL and HrpR/S, using *hrpL*<sub>promoter</sub>:*gfp* and *hrpRS*<sub>promoter</sub>:*gfp* reporter constructs. Based on levels of GFP fluorescence, expression of both *hrpL* and *hrpRS* was significantly reduced in Δ*aauS*, Δ*aauR*, and Δ*aatJ* mutants in response to fructose and aspartate (**Fig 1C and 1D**). We also tested the response of these same mutants to glutamate, another potent T3SS-inducing metabolite, and observed a similar decrease in the expression of *avrRpm1*, *hrpL* and *hrpRS* (**S1 Fig**). Together, these data indicate that the *aat/aau* locus regulates T3SS signaling upstream of *hrpRS* expression in response to acidic amino acids. To confirm these GFP reporter results, we measured the levels

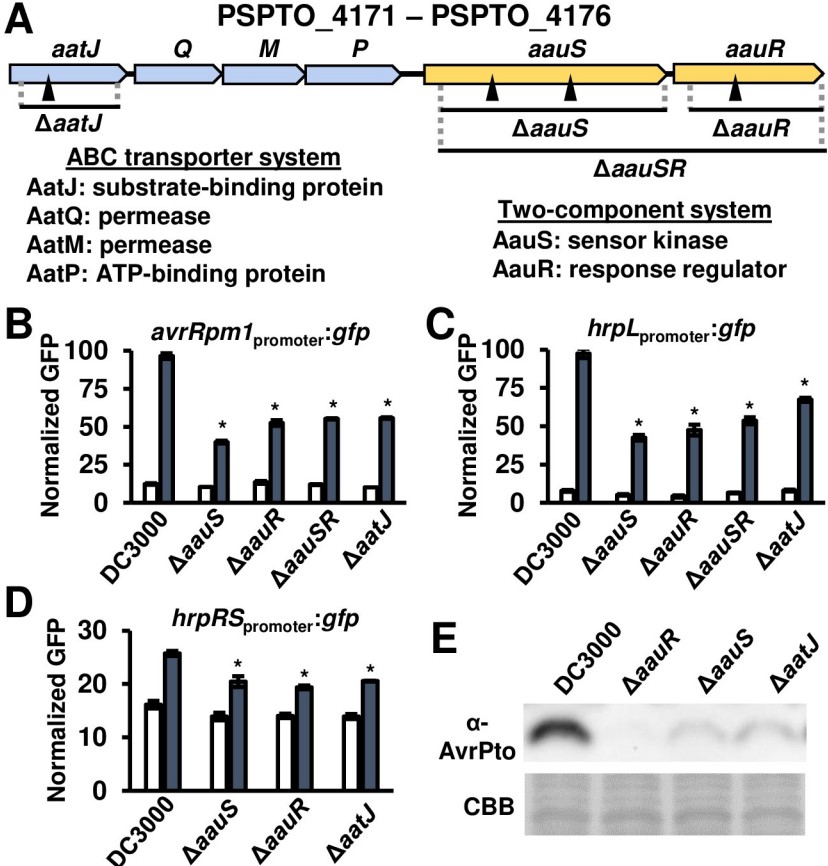

**Fig 1. ABC transporter periplasmic substrate-binding protein AatJ and two-component system AauS-AauR are required for maximum expression of T3SS-associated genes in DC3000. (A)** Schematic of *aat-aau* locus in DC3000. Filled triangles indicate Tn5 insertion sites. **(B-D)** GFP fluorescence of DC3000, Δ*aauS*, Δ*aauR*, Δ*aauSR*, *and* Δ*aatJ*, strains carrying (B) *avrRpm1*$_{promoter}$:*gfp*, (C) *hrpL*$_{promoter}$:*gfp* or (D) *hrpR*$_{promoter}$:*gfp* reporter plasmids. Bacteria were incubated in a minimal medium (MM) with 10 mM fructose (open bars) or 10 mM fructose and 200 μM aspartate (filled bars) for 6 hours (B and C) or 12 hours (D). Graphed are means ± SE of GFP fluorescence normalized to $OD_{600}$ and background fluorescence from empty vector strains; n = 3. Data are representative of three independent experiments. Asterisks denote significant difference based on two-sample *t*-test comparison with DC3000 treated with fructose and aspartate, $P < 0.05$. **(E)** AvrPto levels in DC3000, Δ*aauR*, Δ*aauS* and Δ*aatJ* cultured in MM supplemented with 200 μM aspartate and 10 mM fructose for 4 hours. Upper panel is immunoblot detection of AvrPto in treated bacteria. Lower panel is Coomassie Brilliant Blue (CBB) staining of blot as a loading control.

of endogenous type III effector protein AvrPto in DC3000, Δ*aauS*, Δ*aauR* and Δ*aatJ* strains, and in response to treatment with fructose and aspartate. We observed reduced AvrPto levels in all three mutant strains relative to DC3000 (**Fig 1E**).

To assess if AauS-AauR broadly functions in the perception of T3SS-inducing metabolites [9], we incubated DC3000 and a Δ*aauS*Δ*aauR* strain, both carrying a *hrpL*$_{promoter}$:*gfp* reporter plasmid, in minimal medium supplemented with or without fructose, or supplemented with fructose and citrate, 4-hydroxybenzoic acid (4-hba) or glutamate. Glutamate, in combination with fructose, elicited the highest levels of *hrpL* expression in DC3000 relative to all other metabolite treatments. Citrate and 4hba, in combination with fructose, also induced significantly higher levels of *hrpL* expression in DC3000 compared to fructose alone, albeit to lower levels compared to glutamate and fructose (**S2 Fig**). In response to glutamate, citrate or 4-hba, we measured significantly lower levels of *hrpL* expression in Δ*aauS*Δ*aauR* strain compared to *hrpL* levels in DC3000. However, strikingly, glutamate-induced *hrpL* expression was

attenuated by ~70% in the Δ*aauS*Δ*aauR* strain relative to DC3000 (S2 Fig), whereas *hrpL* expression in Δ*aauS*Δ*aauR* in response to either citrate or 4-hba was only decreased by ~20% relative to DC3000. No significant difference in *hrpL* expression between strains was measured in response to fructose alone (S2 Fig). These data indicate that, among metabolites tested, glutamate is the most potent in terms of its ability to induce *hrpL*, and that loss of AauS-AauR disproportionately effects the response to glutamate compared to other T3SS-inducing metabolites.

## AauS-AauR and AatJ positively regulate T3SS deployment and virulence of DC3000 during infection of Arabidopsis

We next determined if AauS, AauR and AatJ regulate T3SS deployment by DC3000 during plant infection. DC3000, Δ*aauS*, Δ*aauR* and Δ*aatJ* strains were syringe-infiltrated individually into Arabidopsis leaves. Similar to decreased T3SS gene expression observed *in vitro*, we detected decreased AvrPto abundance in all tissues infected with these deletion strains relative to levels of AvrPto in DC3000-infected leaf tissue (Fig 2A). We also infiltrated DC3000 and Δ*aauR* *hrpL*$_{promoter}$:*gfp* reporter strains into Arabidopsis leaves, and measured significantly decreased *hrpL* expression in Δ*aauR*-infected tissue relative to DC3000-infected tissue (Fig 2B). No difference in the population of leaf bacteria was measured at this early time point (S3 Fig). At later time points post-infection, we measured significantly decreased populations of bacteria in Arabidopsis leaves infected with Δ*aauS*, Δ*aauR* and Δ*aatJ* mutants relative to DC3000 (Fig 2C, S3 Fig), as well as observed decreased disease symptoms in Arabidopsis

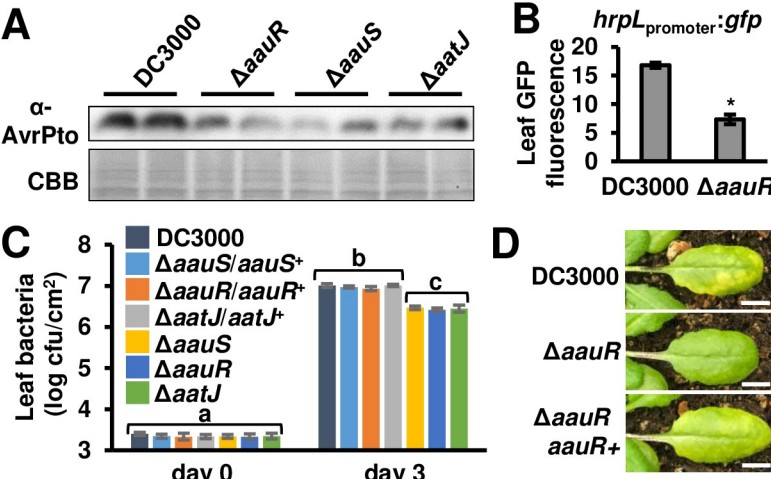

**Fig 2. AatJ, AauS and AauR are required for maximal level of DC3000 virulence in Arabidopsis leaves. (A)** AvrPto levels in Arabidopsis leaves syringe-infiltrated with 1 x 10^8 cfu/mL of DC3000, Δ*aauS*, Δ*aauR*, Δ*aauSR*, or Δ*aatJ* strains. Upper panel is immunoblot detection of AvrPto in infected leaf tissue collected 6 hours post-infiltration. Lower panel is Coomassie Brilliant Blue (CBB) staining of blot as a loading control. **(B)** GFP fluorescence of Arabidopsis leaf tissue syringe-infiltrated with 5 x 10^8 cfu/mL of DC3000 or Δ*aauR* *hrpL*$_{promoter}$:*gfp* reporter strains. Graphed are means ± SE of GFP fluorescence from infected tissue 6 hours post-infection, n = 3. Asterisks denote significant difference between strains based on *t*-test, *P* < 0.001. **(C)** 1 x 10^6 cfu/mL of DC3000, Δ*aauS*, Δ*aauR*, Δ*aatJ* and respective complemented strains were syringe-infiltrated into Arabidopsis leaves. Leaf bacteria populations were enumerated on day 0 and day 3 by serial dilution plating of leaf extracts. Graphed are log-transformed means ± SE of bacteria colony-forming units (cfus) isolated from infected tissue, n = 6 for day 0 and n = 8 for day 3. Small-case letters denote significance groupings based on ANOVA with Bonferroni correction, *P* < 0.01. Data shown were pooled from two independent experiments. Results from additional replicated experiments are shown in S3 Fig. **(D)** Photograph of disease symptoms on leaves 3 days post-syringe infiltration with 1 x 10^6 cfu/mL of DC3000, Δ*aauR* or Δ*aauR*/*aauR*+. White line is 0.5 cm.

leaves infected with the Δ*aauR* mutant (**Fig 2D**). No differences in growth of DC3000, Δ*aauS*, Δ*aauR* or Δ*aatJ* strains cultured in KB medium, LB medium, or a defined minimal medium were observed (**S4 Fig**, **Fig 3D**), indicating that decreased growth in leaf tissue is not due to a general fitness defect of these mutants. Together, these data show that *aauS*, *aauR* and *aatJ* are required in DC3000 virulence and T3SS deployment during infection of Arabidopsis.

## AauS/-R activates expression of *aatJQMP* genes required for maximal uptake of acidic amino acids by DC3000

In the soil bacterium *Pseudomonas putida* KT2440, AauS and AauR regulate the expression of *aatJQMP* in response to extracellular glutamate and aspartate [20–21]. To investigate if *aatJQMP* expression in DC3000 is regulated in a similar manner, we fused the promoter of *aatJ* to a promoterless *gfp* and introduced a plasmid carrying this *aatJ*_promoter:*gfp* reporter into DC3000. We then cultured this reporter strain in minimal medium amended with aspartate or glutamate. We also tested thirteen other amino acids for their ability to induce *aatJ* expression. Consistent with the reported function of *aatJQMP* in *P. putida* KT2440, the highest levels of

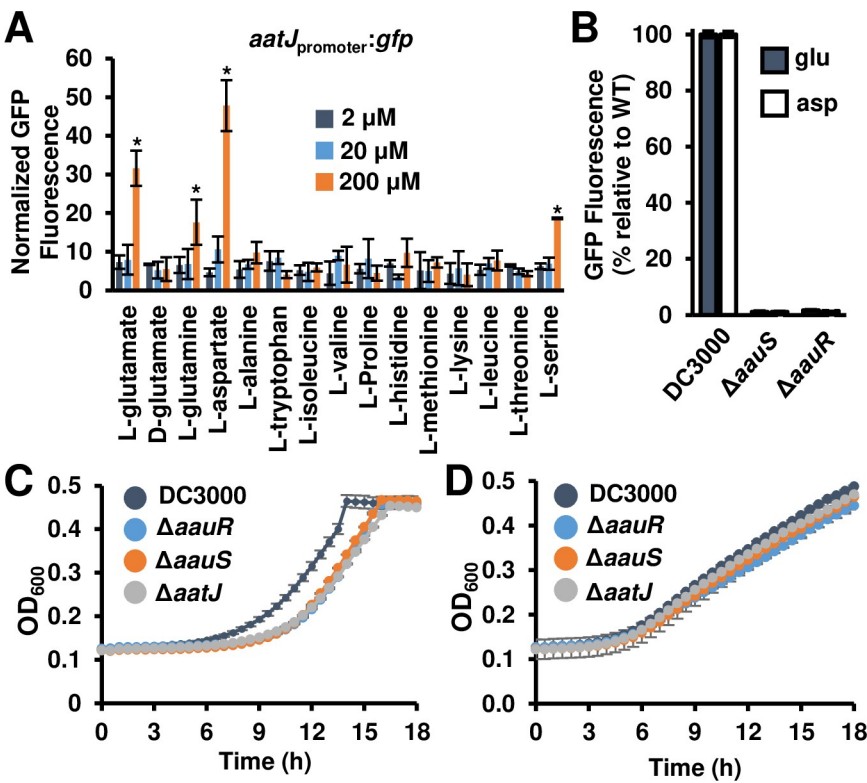

**Fig 3. Genes within the *aat/aau* locus are required for glutamate-induced growth of DC3000. (A)** Expression of the *aatJ promoter* in response to specific amino acids. Graphed is GFP fluorescence from a DC3000 *aatJ*_promoter:*gfp* reporter strain incubated for 24 hours in a minimal medium supplemented with 2, 20 or 200 μM of amino acids as indicated. Graphed are means ± SE of GFP fluorescence normalized to $OD_{600}$ and background fluorescence from empty vector strains; n = 3. Asterisks denote significant difference based on *t*-test comparison to 2 μM treatment condition, $P < 0.05$. **(B)** GFP fluorescence of DC3000, Δ*aauS*, and Δ*aauR aatJ*_promoter:*gfp* reporter strains incubated with 10 mM fructose and 200 μM glutamate or 200 μM aspartate. Graphed are means ± SE of GFP fluorescence normalized to $OD_{600}$ and background fluorescence from empty vector strains; n = 3. **(C-D)** Growth of DC3000, Δ*aatJ*, Δ*aauS*, and Δ*aauR* strains in (C) M9 minimal medium containing 10 mM glutamate as the sole carbon and nitrogen resource, or (D) standard M9 minimal medium containing glucose and ammonium chloride. Cultures were grown for 24 hours at 28˚C. Optical density at λ = 600 nm ($OD_{600}$) of each well was measured at 30 min intervals. Graphed are means ± SD of $OD_{600}$ readings, n = 3. Results in all panels are representative of 3 independent experiments.

induction of *aatJ* expression occurred in response to aspartate and glutamate, with little to no induction of *aatJ* expression by the remaining amino acids tested with the exception of weaker induction by serine and glutamine (**Fig 3A**). We also introduced the *aatJ*<sub>promoter</sub>:*gfp* reporter construct into Δ*aauS* and Δ*aauR* strains and measured essentially no expression of *aatJ* in these mutants when cultured in minimal medium containing glutamate or aspartate (**Fig 3B**). Together, these data show that aspartate and glutamate induce the expression of *aatJQMP* in DC3000 and this response requires both AauS and AauR.

Based on previous functional studies of the *P. putida* Aat ABC transporter [20] and our results above, we predicted that the transporter encoded by *aatJQMP* in DC3000 is necessary for uptake/utilization of acidic amino acids. To test this, we examined the growth rate of DC3000, Δ*aauS*, Δ*aauR* and Δ*aatJ* strains cultured in minimal medium M9 supplemented with glutamate as the sole N and C source. We measured a delay in growth of all three deletion mutants relative to DC3000 (**Fig 3C**). Growth of all three mutant strains could be restored by introduction of plasmids carrying respective wild type alleles of each mutated gene (**S5 Fig**). A similar delay in growth was not observed for Δ*aauS*, Δ*aauR* or Δ*aatJ* strains cultured in minimal medium supplemented with ammonium and glucose as N and C sources, respectively (**Fig 3D**). We conclude from these data that the *aat*/*aau* locus encodes for proteins that contribute to uptake of extracellular acidic amino acids by DC3000.

## Transport and T3SS regulation functions of the *aat*/*aau* locus are genetically separable

*AatQMP* encodes a predicted ABC transporter, with AatM and AatQ forming a membrane-spanning channel, and AatP functioning as an ATP-hydrolyzing protein that provides energy for metabolite transport [20, 22]. To investigate the role of AatMQP transporter function in T3SS deployment, we deleted *aatP* from the DC3000 genome. Consistent with a loss of glutamate transport, the resulting Δ*aatP* mutant had delayed growth in M9 medium with glutamate as a sole N and C source (**Fig 4A**), but no general growth defect in King's B medium (**S6 Fig**) or standard M9 amended with glucose and NH₄Cl (**Fig 4A**). However, the Δ*aatP* mutant had DC3000 levels of *avrRpm1* expression in response to fructose and aspartate (**Fig 4B**), as well as DC3000 levels of *hrpL* expression when infiltrated into Arabidopsis leaves (**Fig 4C**). We also did not observe any decrease in growth of Δ*aatP* mutant in Arabidopsis (**Fig 4D**). Similar phenotypes were observed for a Δ*aatQ*Δ*aatM* mutant (**S7 Fig**). These data provide genetic evidence that the Aat transporter is dispensable for AatJ-AauS-AauR regulation of T3SS induction and virulence but is nonetheless required for growth on glutamate.

## An AauR binding motif upstream of *hrpRS* is required for maximal T3SS deployment and virulence of DC3000 during infection of Arabidopsis

In *P. putida*, AauR regulates the expression of *aatJQMP* by binding to an inverted DNA repeat motif "TTCGG-(N,4)-CCGAA" located upstream of *aatJ* [20]. This motif is also present within the promoter region of *aatJ* in DC3000 (**Fig 5A**). We searched for this AauR-binding motif (Rbm) within the DC3000 genome sequence ([23], *Pseudomonas* genome database), and discovered that an identical Rbm is also present within the intergenic region upstream of *hrpRS* (**Fig 5A**). To assess if this Rbm is required for *hrpRS* expression, we deleted a 20-bp region containing the Rbm from our *hrpRS*<sub>promoter</sub>:*gfp* reporter plasmid and introduced this altered reporter plasmid into DC3000. We also used allelic exchange to delete the same Rbm-containing region upstream of *hrpRS* in the DC3000 chromosome, generating a ΔRbm mutant. In response to fructose and aspartate, DC3000 carrying the *hrpRS*<sub>promoter</sub>(ΔRbm):*gfp* reporter had significantly decreased levels of GFP fluorescence compared to DC3000 carrying an unaltered

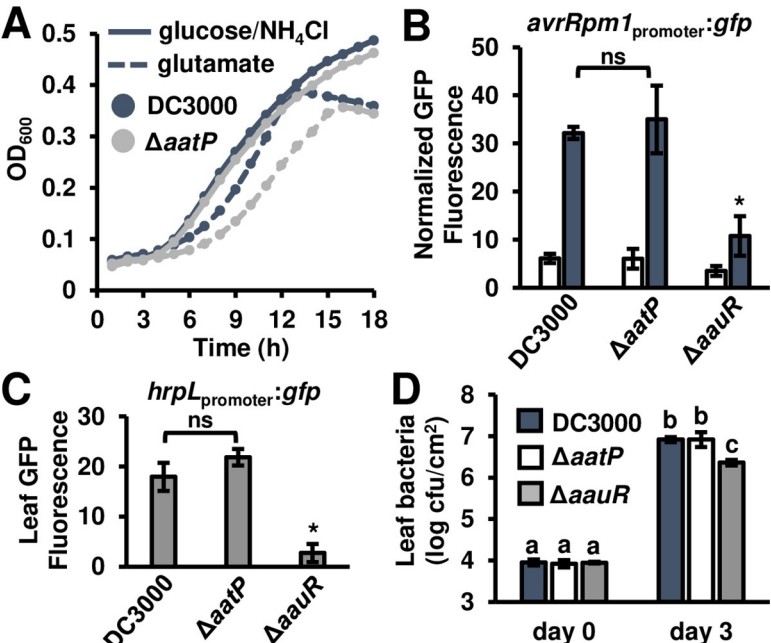

**Fig 4. Amino acid transport and T3SS-inducing functions of the *aat*/*aau* locus are genetically separable. (A)** Growth of DC3000 and Δ*aatP* strains in M9 minimal medium supplemented with 10 mM glutamate. Graphed are means of optical density at λ = 600 nm ($OD_{600}$), n = 3. **(B)** GFP fluorescence from DC3000, Δ*aatP*, and Δ*aauR* *avrRpm1*$_{promoter}$:*gfp* reporter strains cultured in minimal medium (MM) with 10 mM fructose (open bars) or 10 mM fructose and 200 μM glutamate (filled bars) for 6 hours. Graphed are means ± SE of GFP fluorescence normalized to $OD_{600}$ and background fluorescence from empty vector strains; n = 4. Asterisk denotes significant difference based on *t*-test comparison with DC3000, $P < 0.01$; abbreviation ns is not significant, α = 0.05. **(C)** GFP fluorescence of Arabidopsis leaf tissue syringe-infiltrated with 1 x $10^8$ cfu/mL of DC3000, Δ*aatP* or Δ*aauR* strains each carrying a *hrpL*$_{promoter}$:*gfp* reporter plasmid. Graphed are means ± SE of GFP fluorescence from infected tissue 6 hours post-infection, n = 4. Asterisk denotes significant difference based on *t*-test comparison with DC3000, $P < 0.01$; abbreviation ns is not significant, α = 0.05. **(D)** 1 x $10^6$ cfu/mL of DC3000, Δ*aatP* or Δ*aauR* strains were syringe-infiltrated into Arabidopsis leaves. Leaf bacteria populations were enumerated on day 0 and day 3 by serial dilution plating of leaf extracts. Graphed are log-transformed means ± SE of bacteria from infected plants, n = 3 for day 0 and n = 4 for day 3. Small-case letters denote significance groupings based on pairwise *t*-tests, $P < 0.05$. Results in all panels are representative of 3 independent experiments.

*hrpRS*$_{promoter}$:*gfp* reporter (**Fig 5B**). Levels of GFP fluorescence from the *hrpRS*$_{promoter}$(ΔRbm):*gfp* reporter strain were similar to levels from a *hrpRS*$_{promoter}$:*gfp* Δ*aauR* strain, suggesting deletion of Rbm is sufficient to phenocopy loss of *aauR*. We also observed significantly decreased *avrRpm1* and *hrpL* expression in the ΔRbm mutant under the same inducing conditions (**Fig 5C and 5D**).

We next assessed if AauR can directly bind to Rbm upstream of *hrpRS*. We produced a recombinant AauR protein in *E. coli* and tested this protein for interaction with DNA fragments with Rbm-containing *hrpRS* and *aatJ* promoter sequences. We used the bead-based AlphaScreen technology to measure AauR binding to these DNA fragments, with luminescence generated by increased proximity of protein- and DNA-bound beads serving as readout for detecting protein-DNA interactions. Consistent with *P. putida* experiments, we detected binding of AauR to *aatJ* promoter DNA sequence (**S8 Fig**). We also detected significant interaction between AauR and *hrpRS* promoter DNA sequence relative to control assays containing non-biotinylated *hrpRS* promoter DNA (**Fig 5E**). No interaction was detected between AauR and *hrpRS* promoter DNA lacking the Rbm sequence (ΔRbm), confirming the specificity of

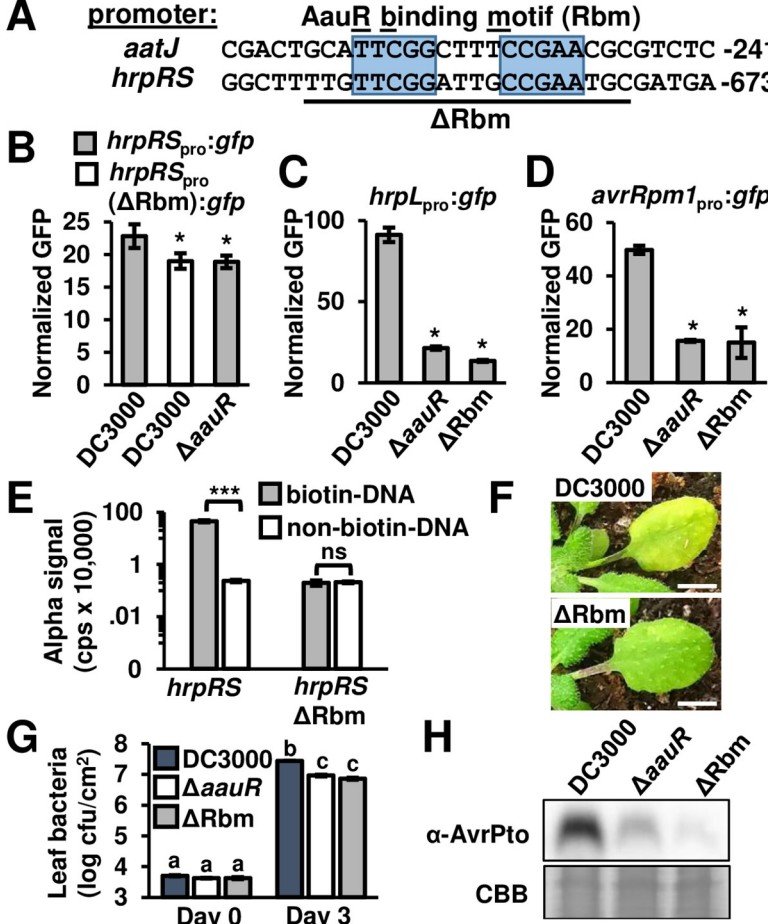

**Fig 5. An AauR binding motif within the *hrpRS* promoter is required for T3SS deployment and maximal virulence of DC3000 in Arabidopsis. (A)** Alignment of DC3000 *aatJ* and *hrpRS* promoter regions containing the AauR-binding motif (Rbm). Conserved palindromic repeats are highlighted in blue. Numbers are distances in base pairs of each Rbm from predicted translation start sites for each gene. The 20 bp region deleted in ΔRbm strain is underlined. **(B-D)** GFP fluorescence from bacteria cultured in minimal medium with 10 mM fructose and 200 uM aspartate for 6 hours. Graphed are means ± SE of GFP fluorescence normalized to $OD_{600}$ and background fluorescence from empty vector strains; n = 3. (B) DC3000 carrying a *hrpRS*<sub>promoter</sub>:*gfp* reporter or *hrpRS*<sub>promoter</sub>(ΔRbm):*gfp* reporter lacking the predicted Rbm. A Δ*aauR* strain carrying *hrpRS*<sub>promoter</sub>:*gfp* was included for comparison. (C) DC3000, Δ*aauR* and ΔRbm strains each carrying a *hrpL*<sub>promoter</sub>:*gfp* reporter. (D) DC3000, Δ*aauR* and ΔRbm strains each carrying an *avrRpm1*<sub>promoter</sub>:*gfp* reporter. Asterisks in B-D denote significant difference between strains based on *t*-test, $P < 0.05$. **(E)** AlphaScreen assay of recombinant AauR protein binding to *hrpRS* promoter DNA with or without Rbm. Graphed are means ± SE of luminescence from assay wells, n = 4. Asterisks indicate $P < 0.001$ based on *t*-test, ns is not significant at α = 0.05. **(F-H)** 1 x $10^6$ cfu/mL of DC3000 or ΔRbm were syringe-infiltrated into Arabidopsis leaves. (F) Photographs of disease symptoms taken 3 days post-infection. White bar is 0.5 cm. (G) Leaf bacteria populations enumerated on day 0 and day 3 by serial dilution plating of leaf extracts. Graphed are log-transformed means ± SE of bacteria colonies isolated from infected tissue, n = 3 for day 0 and n = 3 for day 3. Small-case letters denote significance groupings based on pairwise *t*-tests, $P < 0.001$. (H) AvrPto levels in Arabidopsis leaves syringe-infiltrated with 5 x $10^8$ cfu/mL of DC3000 or ΔRbm. Upper panel is immunoblot detection of AvrPto in infected leaf tissue 6 hours post-infiltration. Lower panel is Coomassie Brilliant Blue (CB) staining of blot as a loading control. Results in panels B-H are representative of 3 independent experiments.

the detected AauR-*hrpRS* interaction (**Fig 5E**). We conclude from these data that AauR can bind directly to the *hrpRS* promoter region containing Rbm.

Next, we infiltrated DC3000, Δ*aauR* and ΔRbm strains into Arabidopsis leaves to determine the requirement of Rbm for virulence. We observed that the ΔRbm strain caused decreased

disease symptoms relative to DC3000 (**Fig 5F**). Levels of ΔRbm bacteria in infected tissue were significantly reduced compared to DC3000, yet indistinguishable from Δ*aauR* (**Fig 5G**). The abundance of AvrPto protein produced by the ΔRbm mutant was also lower than DC3000 in infected leaf tissues (**Fig 5H**). These results show that the AauR-binding site in the *hrpRS* promoter is required for maximal expression of T3SS-associated genes and for full virulence and bacterial growth of DC3000 in Arabidopsis.

### Phylogenetics suggests AauS-AauR regulation of *hrpRS* is highly conserved amongst all *P. syringae* with a canonical T3SS

We hypothesized that Rbm might be variable or even absent in some strains as a consequence of adaptation of *P. syringae* to particular plant hosts or host micro-environments. To investigate, we characterized 38 genome-sequenced isolates of *P. syringae* representing nine phylogroups with strains that carry a canonical T3SS tripartite pathogenicity island (T-PAI) [24]. We first confirmed the reported phylogroup associations of each representative strain by generating a *citrate synthase* (*cts*)—based phylogeny that included 800 additional *P. syringae cts* sequences from genomes available at NCBI [25]. We then extracted and aligned nucleotide sequences between *hrpH* and *hrpR* from genome sequences of all 38 isolates (**S9 Fig**). This alignment showed that the Rbm motif (TTCGG-(N,4)-CCGAA) is absolutely conserved (**Fig 6A and 6B**, **S9 Fig**), with the exception of a single nucleotide polymorphism in the Rbm sequence of *P. syringae* pv. *phaseolicola* strain 1448a. Guanine and thymine nucleotides that

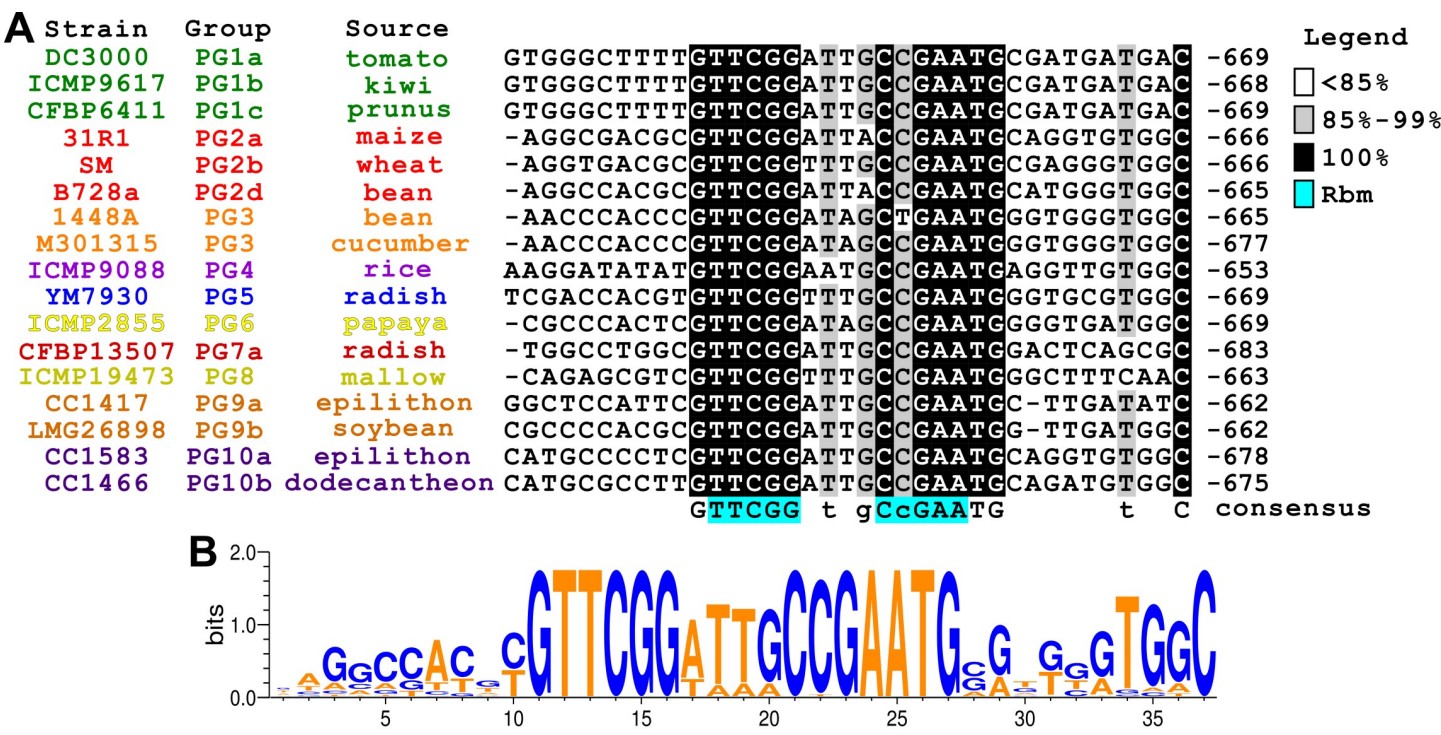

**Fig 6. An AauR binding motif upstream of *hrpRS* is highly conserved in all *P. syringae* phylogroups that possess a canonical type III secretion system. (A)** Alignment of *hrpRS* promoter sequences from 17 strains chosen to represent 9 phylogroups (PG) of *P. syringae* that carry a canonical T3SS. Strains are color coded according to phylogroup. Numbers correspond to nucleotide position relative to predicted *hrpR* translation start sites. Letters below alignment are consensus sequence of nucleotides based on >90% identity. Highlighted in blue is AauR-binding motif (Rbm) from *aatJ* promoter. **(B)** Sequence logo generated from multiple sequence alignments of *hrpRS* promoter region containing Rbm from 38 *P. syringae* isolates. Relative sizes and heights of the letters indicate their frequency in the sequences and information content, respectively. AT and GC content is highlighted by orange and blue coloring, respectively.

immediately flank the Rbm are also highly conserved, as well as an "ATTG"-rich signature within the spacer sequence between inverted repeats (**Fig 6B**). Sequences farther upstream and downstream of the Rbm are less conserved (**Fig 6A**, **S9 Fig**). Nevertheless, these regions are more conserved within phylogroups than between phylogroups (**Fig 6A**, **S9 Fig**), suggesting vertical inheritance of regions flanking the Rbm. We further investigated inheritance of the *hrpR* region by aligning the *hrpR* open reading frames, along with the intergenic region between *hrpH* and *hrpR*, from all 38 isolates examined. Phylogenies based on these alignments showed strong intra-phylogroup cohesion (**S10 Fig**, **S11 Fig**). Together, these data do not support our hypothesis of positive selection of Rbm between isolates. Rather, they suggest negative selection has occurred to maintain the Rbm within the context of a *hrpRS* promoter region that has been vertically inherited in all T-PAI-carrying isolates.

## AauR is required for maximal T3SS deployment and virulence of bean pathogen *P. syringae* pv. *syringae* B728a

The *aat/aau* locus is conserved among pseudomonads [20, 21]. A phylogenetic analysis of orthologous *aatJ* and AauS sequences from diverse *P. syringae* isolates confirmed vertical inheritance of this locus (**S12 Fig**, **S13 Fig**). Because *aat/aau* genes and the Rbm upstream of *hrpRS* are both conserved, we predicted that all *P. syringae* use the AatJ-AauS-AauR signaling pathway to regulate T3SS-encoding genes in response to host signals. To test this prediction, we deleted the orthologous *aauR* gene in *P. syringae* pv. *syringae* B728a, a pathogen that causes brown spot of bean and belongs to a phylogroup distinct from that of DC3000 [24–26]. Similar to DC3000 Δ*aauR*, B728a Δ*aauR* showed delayed growth in M9 medium containing glutamate as the sole N and C source (**Fig 7A**), but not in standard M9 or LB medium (**S14 Fig**), indicating that AauR is required for the uptake of acidic amino acids in B728a. We then introduced the *hrpL*$_{promoter}$:*gfp* reporter plasmid into B728a and B728a Δ*aauR* strains. With the B728a reporter strain, we detected a significant increase in *hrpL* expression in response to fructose and aspartate relative to fructose only, indicating that aspartate is also an inducer of T3SS genes in B728a (**Fig 7B**). Consistent with our prediction of conserved AauS-AauR function, *hrpL* expression was significantly decreased in B728a Δ*aauR* cultured with fructose and aspartate (**Fig 7B**). We also syringe-infiltrated B728a and B728a Δ*aauR* into leaves of *Phaseolus vulgaris* (common bean), and measured significantly reduced bacterial populations (**Fig 7C**) in leaves infected with B728a Δ*aauR*. We conclude from these data that AauR is required for aspartate-induced expression of T3SS genes in B728a, and is required for maximal growth of B728a during infection of host bean plants.

## Discussion

The evolutionary transition of bacteria from non-pathogenic to pathogenic often involves the horizontal acquisition of genes that confer novel virulence traits [27]. However, acquiring virulence-promoting genes is likely often only a "foothold" moment, as additional genomic changes must occur to optimize expression and reduce genetic conflicts to maximize fitness gains and reduce fitness costs [28]. Previous phylogenetic analyses revealed that the canonical T3SS-encoding pathogenicity island was gained early in *P. syringae* evolution [29–30]; yet how T3SS-encoding genes within the island became integrated into existing gene regulatory networks, and how these genes became responsive to host signals, was not known. In this study we present evidence that AauS-AauR and AatJ, proteins involved in regulating the housekeeping task of acidic amino acid uptake in both non-pathogenic and pathogenic pseudomonads, were co-opted to regulate T3SS deployment in *P. syringae* strain DC3000 (**S15 Fig**). Our phylogenetic analyses indicate that co-option of AatJ-AauS-AauR signaling, through acquisition of

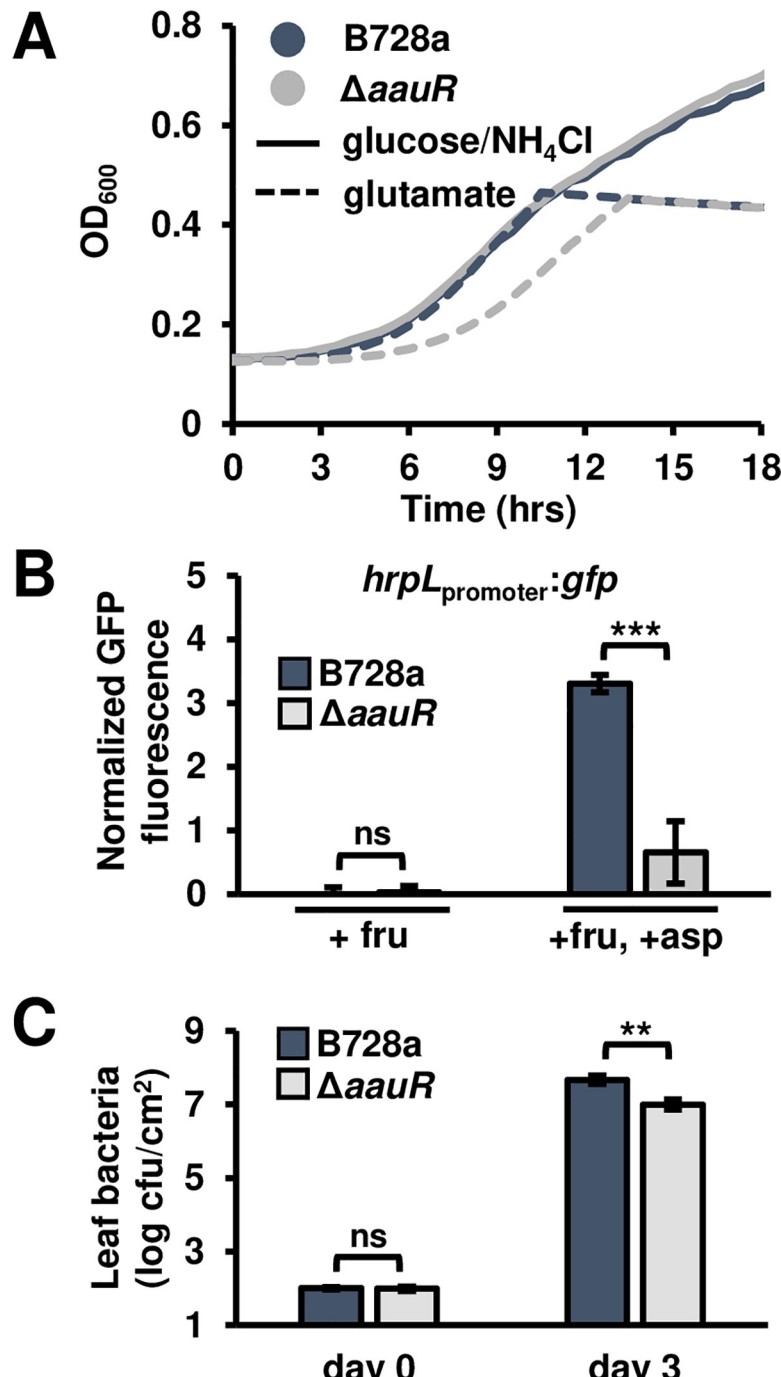

**Fig 7. AauR is required for glutamate-dependent growth, T3SS deployment and virulence of the bean pathogen *P. syringae* pv. *syringae* B278a. (A)** Growth of wild type B728a and B728a Δ*aauR* in M9 minimal medium supplemented with 10 mM glutamate as the sole carbon and nitrogen source. Cultures were inoculated at an optical density at λ = 600 nm (OD$_{600}$) of 0.1 and grown for 24 hours at 28˚C within a 96-well plate. Graphed are means ± SD of OD$_{600}$ readings taken at 30 min intervals, n = 3. **(B)** GFP fluorescence of B728a and B728a Δ*aauR* strains carrying *hrpL*$_{promoter}$:*gfp* reporter plasmids. Bacteria were incubated in a minimal medium (MM) with 10 mM fructose (fru) or 10 mM fructose and 200 μM aspartate (asp) for 6 hours. Graphed are means ± SE of GFP fluorescence normalized to OD$_{600}$ and background fluorescence from empty vector strains; n = 3. **(C)** Leaves of 14-day-old bean plants were syringe-infiltrated with 1 x 10$^{4}$ cfu/mL of B728a. Bacterial populations in infected leaves 0 and 3 days post-infection. Graphed are log-transformed means ± SE of colony-forming units (cfus) in infected leaf tissue, n = 4. Data in all panels are representative of results from three independent experiments. Asterisks denote significant differences based on *t*-test. ** is $P < 0.05$, *** is $P < 0.01$ and ns is not significant at α = 0.05.

an Rbm upstream of *hrpRS*, occurred early in *P. syringae* evolution and was maintained throughout subsequent diversification of the species. The placement of an Rbm upstream of *hrpRS* may have occurred through recombination events that captured and re-purposed existing Rbm elements within the genome, or by *de novo* assembly of an Rbm through the accumulation of random mutations [31]. Alternatively, AauS-AauR regulation of *hrpRS* may pre-date *P. syringae* speciation, with an Rbm already present within the *hrpRS* promoter upon integration of the T3SS pathogenicity island into the ancestral *P. syringae* genome.

Co-option of environment-responsive signaling pathways for virulence has occurred in other pathogenic bacteria. In *Salmonella enterica*, the $Mg^{2+}$-responsive two-component system PhoQ-PhoR regulates the expression of multiple virulence-associated genes, including T3SS-associated genes within the SPI-2 pathogenicity island, as well as numerous non-virulence genes scattered throughout the genome [32]. The widespread presence and conserved functions of PhoQ-PhoR in both pathogenic and non-pathogenic Gram-negative bacteria suggest it was likely present in an ancestral strain of *Salmonella* and adapted for virulence [33]. In the gall-forming plant pathogen *Agrobacterium tumefaciens*, virulence (*vir*) genes are regulated by VirA-VirG, a two-component system activated by plant-derived phenolic compounds [4]. Interestingly, activation of VirA is enhanced by host-derived sugars through the action of Chromosomal virulence gene E (ChvE), a periplasm-localized SBP with broad specificity for binding aldose monosaccharides [34–36]. Similar to the dual functions of AatJ in virulence and transport, ChvE also functions as an SBP for MmsAB-AraD, a sugar-specific ABC transporter that is dispensable for *Agrobacterium* virulence [36]. The striking parallels between AatJ and ChvE indicate that two unrelated plant pathogens have similarly co-opted the metabolite-binding capabilities of SBPs, suggesting this mechanism of virulence gene regulation may be commonplace.

AauS is a predicted extracytoplasmic sensor kinase with an N-terminal periplasmic domain and a C-terminal intracellular histidine kinase domain. Based on well-established models of two-component system activation [37], as well as previous functional characterization of the *aat/aau* locus in *P. putida* [20, 21], the presence of aspartate and glutamate in the periplasm likely causes AauS to auto-phosphorylate, followed by *trans*-phosphorylation and activation of AauR. Multiple mechanisms of AauS activation by aspartate and glutamate are possible. The N-terminus of AauS possesses a predicted CACHE (CA2+ channels, chemotaxis receptors) domain involved in small molecule binding (Pfam.xfam.org, [38]), and has structural similarity to C4-dicarboxylic transport B (DctB) (Phyre2, [39]), a sensor kinase in Rhizobia that recognizes plant-derived dicarboxylates [40, 41]. Based on this similarity to DctB, aspartate and glutamate may directly bind to the periplasmic domain of AauS to activate this pathway. Alternatively, AauS may perceive aspartate and glutamate indirectly through interactions with AatJ. In this regard, ligand-bound SBPs can function in signal transduction by directly interacting with and activating an associated chemotaxis receptor or sensor kinase [34, 42–44]. Therefore, AauS may recognize an AatJ-metabolite complex. This latter model is consistent with our observation that *aatJ* is required for maximal induction of *hrpRS* by aspartate and glutamate. Additional experiments to assess metabolite binding to AatJ and AauS, as well as *in vivo* studies of possible AatJ-AauS interactions, will be necessary to differentiate between these possible activation mechanisms.

Our observation that T3SS deployment by Δ*aatJ*, Δ*aatS* and Δ*aauR* strains is reduced *in planta* suggests that aspartate and/or glutamate are key virulence-inducing signals for DC3000 within the leaf apoplast. Furthermore, the conserved nature of Rbm upstream of *hrpRS* suggests that essentially all pathogenic strains of *P. syringae* use AauS-AauR to regulate their virulence. This possibility is supported by our evidence that *hrpL* expression was induced by aspartate in strain B728a, and that *aauR* was required for maximal growth of strain B728a in

bean. Similar to targeting of PAMPs by PRRs, we reason that *P. syringae* may use aspartate and glutamate as virulence-inducing cues because these metabolites are abundant in plant tissues and are essential for plant viability. Across a broad phylogeny of plant species, aspartate and glutamate are the most abundant amino acids in plants [45]. At the cellular level, both metabolites play central roles in amino acid biogenesis and catabolism pathways, including the assimilation of nitrogen into organic compounds [46, 47]. At the whole plant level, aspartate and glutamate participate in nitrogen transport from roots to shoots, and between source and sink tissues [48]. Glutamate may also function as a trigger for long-distance wounding signals in plants [49]. Within the leaf apoplast, the steady state levels of extracellular metabolites likely reflect the combined rates of uptake and release from vascular tissue as well as from local mesophyll cells. Interestingly, aspartate and pyroglutamic acid, a derivative of glutamate, were both decreased in exudates from the defense-heightened Arabidopsis *mkp1* mutant [9], suggesting exudation of acidic amino acids from plants may be defense-regulated. In addition to regulating T3SS-encoding genes within the apoplast, metabolites that leach onto the leaf surface may serve as virulence signals. Recently, the aspartate- and glutamate-specific chemotaxis receptor PscA in DC3000 was shown to be required for maximal virulence of bacteria inoculated onto the surface of tomato leaves [50], indicating aspartate and glutamate may also be important chemotactic signals during motile stages of infection on the leaf surface.

Identifying AauR as a direct regulator of *hrpRS* is an important step towards addressing how the HrpR-HrpS-HrpL signaling module is activated during infection. However, aspartate- and glutamate-dependent induction of this pathway was only partially abrogated in Δ*aatJ*, Δ*aatS* and Δ*aauR* mutants, indicating that additional Aat/Aau-independent signaling pathways are involved in perceiving these metabolites. Furthermore, loss of AauS-AauR had little to no impact on *hrpL* expression in DC3000 in response to fructose, citrate, or 4-hydroxybenzoic acid. Therefore, these metabolites must also induce T3SS-associated genes through AauS/AauR-independent mechanisms. Collectively, these data suggest that regulation of type III secretion in *P. syringae* is complex, with contributions from redundant pathways that respond to the same metabolite, as well as from pathways that function independently in response to distinct metabolic signals. Previously, the two-component system RhpS/RhpR (Regulator of *hrp* genes, Sensor and Regulator) was identified as a regulator of *hrpRS* expression in *P. savastanoi* and *P. syringae* [17]. RhpR binds to the *hrpRS* promoter in a region located approx. 300 nucleotides further upstream from the Rbm [51]. In *rhpS⁻* mutants, RhpR constitutively represses *hrpRS* expression, resulting in decreased T3SS deployment and reduced virulence [17]. However, mutants lacking both *rhpS* and *rhpR* have wild-type levels of *hrpRS* expression and are fully virulent, indicating that RhpS/RhpR does not directly regulate T3SS genes in response to host-specific signals [17]. We hypothesize that additional host metabolite-responsive two-component systems, apart from AauS-AauR, may provide signaling inputs. Alternatively, metabolite-mediated regulation of T3SS may also occur intracellularly. In this regard, we recently identified a predicted metabolite-sensing transcription regulator SetA that is genetically required for sugar-induced expression of *hrpL* [18]. Because both sugars and amino acids are present within the leaf apoplast, AauS/AauR- and SetA-dependent host perception mechanisms likely function simultaneously during infection. How T3SS induction by different classes of host metabolites may be coordinately regulated in *P. syringae* is an open question for future investigation.

## Materials and methods

### Bacterial strains and growth conditions

Bacterial strains and plasmids are listed in **S1 Table**. *Pseudomonas syringae* were cultured in a modified King's B (KB) [52] medium (1% (w/v) peptone, 1% tryptone, 0.1% MgSO$_4$-7H$_2$O,

0.1% $K_2HPO_4$, and 1% (v/v) glycerol). For *in vitro* GFP reporter assays, *P. syringae* were cultured in a modified *hrp*-inducing minimal medium (MM, [9]) or M9 medium [53] amended with appropriate carbon sources. Aspartate, glutamate, citrate and 4-hydroxybenzoic acid were prepared as 20 mM stocks in water, 0.2 μm filtered and stored at -20°C. Fructose was prepared as a 1 M stock in water, 0.2 μm filtered and stored at room temperature. *Escherichia coli* were cultured in Luria-Bertani (LB) [54] broth at 28°C. Prior to *aatJ* promoter activity experiments, *P. syringae* were cultured at $OD_{600} = 5.0$ in standard M9 medium at 28°C with shaking at 200 rpm for four weeks, with bacteria collected by centrifugation and resuspended in fresh M9 medium every two days. For all other assays *P. syringae* were cultured for two days on KB agar (1.5%) at room temperature prior to use.

## Tn5 transposon mutagenesis screen

DC3000 carrying an *avrRpm1*$_{promoter}$:*gfp*::pBBR1-MCS2 construct [55,56] were mutagenized by transposon Tn*5* through conjugative transfer of suicide vector pUT:mini-Tn*5* Sm/Sp as described previously [18]. Briefly, to select for Tn*5*$^+$ colonies, conjugation mixtures were plated onto the surface of a nitrocellulose filter placed on the surface of KB agar (1.5%) containing spectinomycin (150 μg/mL), rifampicin (50 μg/mL) and kanamycin (30 μg/mL). The plate was kept at room temperature for three days until colonies became visible. To remove residual KB medium, the nitrocellulose membrane was transferred to 1.5% agar solidified with water. After 24 hours, the membrane was transferred to 1.5% agar plates containing 50 mM fructose and 5 mM aspartate. After six hours, GFP fluorescence of colonies was visually scored using a Leica MZFLIII stereomicroscope. Approximately 20,000 Tn*5*$^+$ colonies were screened in this manner, and 400 selected for further study. As a second round of screening, a Tecan Spark 10M plate reader was used to measure the GFP fluorescence of each mutant strain cultured at a starting $OD_{600} = 0.1$ for eight hours in liquid MM supplemented with 50 mM fructose and 1 mM aspartate. Approximately two hundred mutant strains with a >20% reduction in GFP fluorescence relative to a wild type DC3000 culture were selected, and Tn*5* insertion sites identified by Illumina sequencing of genomic DNA as described previously [18].

## Construction of *P. syringae* deletion mutants

Strains of *P. syringae* with deletions of *aauS*, *aauR*, *aatJ*, *aatQM*, *aatP*, *aauSR* or the *aauR* binding motif (Rbm) were generated by suicide vector-mediated allelic exchange. DNA fragments flanking each targeted gene were PCR amplified from DC3000 genomic DNA, fused into a single PCR product by a second round of overlapping PCR, and subcloned into suicide vector pEX18Km [57]. Deletion constructs were introduced into *P. syringae* by tri-parental mating. Briefly, 2 mL overnight cultures of *E. coli* containing either the pEX18Km deletion construct or helper plasmid pRK600 [58] were incubated at 28°C with a shaking at 200 rpm. The overnight cultures were washed with sterile water, mixed at a volumetric ratio of 1:1:10 (*E. coli* pRK600: *E.coli* pEX18Km: *P. syringae*), and spread onto a KB agar plate without antibiotics. After incubating overnight at 28°C, the conjugation mixtures were resuspended in sterile water and plated onto KB agar amended 50 μg/mL rifampicin and 30 μg/mL kanamycin to select for the first recombination event. The resulted merodiploid colonies were then plated on KB agar supplemented with 15% v/v sucrose for two days to counter-select for presence of *sacB* and identify deletion mutants. Gene deletions were confirmed by sensitivity of mutant strains to kanamycin and by PCR genotyping using oligonucleotides that anneal to regions that flank the targeted genes. A list of all deletion constructs generated is provided in **S1 Table**, and sequences of oligonucleotides used for vector construction are provided in **S2 Table**.

## Construction of transcriptional reporter plasmids and gene expression plasmids

Reporter plasmids *avrRpm1*$_{promoter}$:*gfp*::pBBR1-MCS2 and *hrpL*$_{promoter}$:*gfp*::pProbe-NT were described previously [59]. To make the *hrpRS*$_{promoter}$:*gfp* reporter, oligonucleotides hrpR-F1_pro and hrpR-1a were used to PCR amplify a 1222-bp DNA fragment containing the *hrpR* promoter region from DC30000 genomic DNA. The resulting PCR fragment was subcloned into BamHI and KpnI sites of pPROBE-NT [60]. The resulted plasmid was named phrpRS$_{promoter}$-gfp. To delete the AauR-binding motif (Rbm) in phrpRS$_{promoter}$-gfp, oligonucleotide pairs hrpR-F1/ hrpR-R7-OVLP and hrpR-R1/hrpR-F6-OVLP were used to PCR amplify two DNA fragments flanking the Rbm site. The resulting DNA fragments were fused together by overlapping PCR using oligonucleotides hrpR-F1-pro and hrpR-R1-pro. NEBuilder HiFi mix (NEB) was used to ligate the resulting PCR product into BamHI/KpnI-digested pPROBE-NT. To make the *aatJ*$_{promoter}$:*gfp* construct, oligonucleotides aatJ_f2 and aatJ_R2a were used to PCR amplify a 506-bp DNA fragment from DC3000 genomic DNA. The resulting PCR fragment was digested with HindIII and EcoRI, and ligated into pPROBE-NT. All constructs used for gene complementation were made using broad host range vector pME6010 [61]. To prepare a linearized vector for subcloning, oligonucleotides pairs pME6010-F1/pME6010-R1 and pME6010-F2/ pME6010-R2 were used to PCR amplify two DNA fragments 2.8 kb and 5.5 kb in length from pME6010. Gene-specific oligonucleotides were used to PCR amplify *aauS*, *aauR* and *aatJ*-containing DNA fragments from DC3000 genomic DNA. NEBuilder HiFi mix (NEB) was used to assemble the resulting PCR products into an intact plasmid. Correct assembly of plasmids was confirmed by PCR and Sanger sequencing. All constructs were confirmed by PCR genotyping and Sanger sequencing. Sequences of oligonucleotides used for constructing plasmids are provided in **S2 Table**.

## *In vitro* and *in planta* GFP transcriptional reporter assays

To measure GFP fluorescence of *P. syringae* cultured in T3SS-inducing media, bacteria were scraped from the surface of KB agar, resuspended in 1 mL of $H_2O$ and washed twice with 1 mL of $H_2O$ using a centrifuge to pellet bacteria between washes. After washing, the suspensions of bacteria were adjusted to an optical density at $\lambda = 600$ nm ($OD_{600}$) of 1.0 and inoculated into wells of black μClear 96-well plates (Greiner Bio-One) containing 90 μL of MM amended with 200 μM of aspartate, glutamate, citrate, 4-hydroxybenzoic acid and/or 10 mM fructose. The final density of bacteria in each well was $OD_{600} = 0.1$. A Spark 10M plate reader (Tecan) was used to measure GFP fluorescence and $OD_{600}$ of each well. Prior to measurements, the plate was incubated in the plate reader at 25°C or ambient (if higher) and shaken at 216 rpm. A humidity cassette (Tecan) was used to prevent sample evaporation. Fluorescence from each well was measured using excitation and emission wavelengths of $485_{nm}$ and $535_{nm}$, respectively, with manual gain set to 100 and settle time set to 1000 ms. $OD_{600}$ was measured using default settings with a 1000 ms settle time. Each fluorescence measurement was divided by the corresponding $OD_{600}$ of each well to determine a culture density-normalized GFP level. Further, culture density-normalized fluorescence values from wells containing empty pPROBE-NT control strains were used for background correction to calculate final normalized GFP fluorescence values. All normalized fluorescence values were divided by 10,000 to simplify graphing of data. Measurements of *hrpL*$_{promoter}$:*gfp* activity in leaf tissue and immunoblot detection of AvrPto were done as described previously [18,59].

## Measurements of *P. syringae* growth in culture

For each culture, 10 μL of *P. syringae* suspended in sterile water was added to 90 μL of either KB broth, M9 medium containing 0.4% glucose and 0.1% ammonium chloride, or a modified

M9 medium lacking glucose/$NH_4Cl$ and supplemented with 10 mM glutamate. Cultures were maintained in wells of a 96-well plate incubated at 28˚C and shaken at 216 rpm. A Spark 10M plate reader was used to measure the $OD_{600}$ of each well every 30–60 minutes using default settings.

## Measurements of *P. syringae* growth *in planta*

Arabidopsis Col-0 seeds were surface sterilized, stratified and plated onto MS agar as described previously [9]. After two weeks, Arabidopsis seedlings were transferred to Sunshine mix soil. Seeds of *Phaseolus vulgaris* (bush bean) cultivar Blue Lake seeds (Ferry-Morse Seed) were germinated directly in Sunshine mix. All plants were maintained at 22˚C in a 10-hour-day growth chamber. For Arabidopsis infection, inoculums of *P. syringae* in water were adjusted to $OD_{600}$ = 0.002, then infiltrated into fully expanded leaves of five-week-old plants. For bean infections, inoculums were adjusted to $OD_{600}$ = 0.001, then infiltrated into leaves of five-week-old plants. The infected plants were kept at 22˚C in 10-hour-day growth chamber. Bacterial populations in infected leaves were measured as previously described [18]. Each sample is bacteria isolated from three leaf disks taken from three leaves of a single plant.

## Expression and purification of recombinant AauR

Recombinant AauR with a C-terminal 6xHis epitope tag was produced in *E. coli*. To prepare an expression construct, oligonucleotides *aauR_28a*_F and *aauR_28a*_R were used to PCR-amplify *aauR* from DC3000 genomic DNA, and oligonucleotides *pET28a*_F and *pET28a*_R were used to amplify a linearized pET28a fragment. NEBuilder HiFi reaction mix (NEB) was used to assemble the resulting *aauR* and pET28a DNA fragments into an intact plasmid. Correct assembly of plasmids was confirmed by PCR and Sanger sequencing. Sequences of oligonucleotides used for plasmid construction are provided in S2 Table. For protein expression, *E. coli* strain BL21(DE3)(pLysS) harboring the pET28a::*aauR* plasmid were cultured overnight at 37˚C, transferred to 200 mL of LB broth and cultured at 37˚C until $OD_{600}$ of the culture reached 0.6–0.8. AauR expression was induced by addition of IPTG to a final concentration of 0.2 mM. After incubating at 20˚C for 16 hrs, cells were pelleted by centrifugation. The cell pellet was resuspended into 5 mL of lysis buffer (25 mM Tris-HCl pH 7.5, 500 mM NaCl, 20 mM imidazole) and sonicated. The total cell lysate was then centrifuged at 12,000 x *g* for 30 min at 4˚C. The resulting supernatant was added to pre-equilibrated Profinity IMAC Ni-charged resin (Bio-Rad), and the mixture gently mixed by shaking at 4˚C for 1 hr. After loading the mixture onto a column, the resin was washed with 10 column volumes of wash buffer (2.5 mM Tris-HCl pH 7.5, 500 mM NaCl, 50 mM Imidazole). The bound protein was eluted with elution buffer (25 mM Tris-HCl p H 7.5, 300 mM NaCl, 200 mM Imidazole). Elution fractions were analyzed by SDS-PAGE and AauR abundance quantified using Pierce BCA Protein Assay Kit.

## *In vitro* detection of AauR-DNA interactions

The AlphaScreen histidine (nickel chelate) detection kit (Perkin Elmer) was used to detect interactions between recombinant AauR and DNA. Single-stranded biotinylated and non-biotinylated DNA oligonucleotides (Integrated DNA Technologies) were reconstituted into dsDNA probes using the method described in [62]. Sequences of oligonucleotides are provided in S2 Table. Interactions between AauR and DNA were assayed in wells of half-volume white 96-well plates (Greiner Bio-One). In each well, 13 μL containing 1x kit buffer, 0.01% Tween 20, 2% BSA was mixed with 2 μL of 6xHis tagged AauR (in 300 nM-1000 nM range) and 2 μL of 625 nM biotinylated or non-biotinylated dsDNA probe. After incubating at room

temperature for 2 hr or 4°C overnight, 4 μL of 125 μg/ml acceptor beads was added to each well. The mixture was incubated at room temperature for 1 hr, then 4 μL of 125 μg/ml donor beads solution added to each well. After the addition of donor beads, the plate was kept in the dark at room temperature for 1 hr before measuring luminescence from each well using a Bio-Tek Synergy 2 plate reader. All reactions were performed in triplicate and results were successfully repeated with independent batches of purified AauR protein.

### Bioinformatics and phylogenetic analyses

Genome sequences and locus identifiers of *P. syringae* strains analyzed are listed in **S3 Table**. Strains were chosen to encompass all diversity of the canonical type III secretion system, with strains differentiated by phylogroup using the citrate synthase (*cts*) partial coding sequence [24,25]. All genome sequences in NCBI listed under the 'Pseudomonas syringae group' (taxid 136849) were downloaded (May 20, 2019). The *cts* gene sequence was extracted from BLASTN + (v 2.7.1) search results using the *P. syringae* DC3000 *cts* gene sequence as the query, and the complete genome sequences of all other strains as the target BLAST database [63]. The DC3000 *cts* gene sequence and other reference phylogroup *cts* gene sequences were obtained from Supplementary File S1 from [24]. Complete genomes were chosen, when available, for each phylogroup. When possible, potential intra-phylogroup diversity was also captured in the strains selected for analysis based on the *cts* phylogeny; general quality of genome assembly was estimated by comparing total numbers of contigs and total assembly length. Individual representatives from each phylogroup were selected to present in Fig 6A; these strains are listed in **S3 Table**. The amino acid sequence from *P. syringae* DC3000 for HrpR (PSPTO_1379), AatJ (PSPTO_4171), and AauS (PSPTO_4175) were used as queries for BLASTP+ (v 2.7.1) searches against the predicted proteomes of all other genomes [63]. BLASTP hits were manually filtered using empirically derived percent identity cut-offs. Nucleotide sequences for these protein coding regions were used in the phylogenetic tree reconstruction; amino acid sequences for AauS were used as they provided sufficient resolution of intra-phylogroup relationships. The intergenic region upstream *hrpR*, including the predicted AauR binding motif, was extracted using BioPython (v 1.70) [64]; the aligned motif was taken as a subset from the full alignment of the intergenic region. We used MAFFT (v 7.402) specifying the G-INS-i algorithm to align the sequences, and IQ-TREE (v 1.6.9) to generate the maximum likelihood phylogenetic trees; bootstrap supports were computed using both ultrafast bootstraps (-bb 1000) and the SH-aLRT test (-alrt 1000) [65–68]. Trees were visualized using the itol web server v4 (https://itol.embl.de/, [69]). The aligned AauR binding motif was visualized using the texshade (v 1.25) package in TeX Live 2019 (https://www.ctan.org/pkg/texshade;https://www.tug.org/texlive/index.html). Trees and alignments were annotated using Inkscape v0.92 (https://inkscape.org/).

### Statistics

All statistical analyses were done using Minitab, R and Excel software.

### Supporting information

**S1 Fig. DC3000 genes *aauS*, *aauR* and *aatJ* are required for maximal expression of T3SS genes in response to glutamate.** GFP fluorescence of DC3000, Δ*aatJ*, Δ*aauS*, Δ*aauR* and Δ*aauS*Δ*aauR* strains carrying **(A)** *avrRpm1*$_{promoter}$:*gfp*, **(B)** *hrpL*$_{promoter}$:*gfp* or **(C)** *hrpRS*$_{promoter}$:*gfp* reporter plasmids. Bacteria were cultured in a minimal medium (MM) with 10 mM fructose (open bars) and 10 mM fructose plus 200 μM glutamate (filled bars) for 6 hours. Graphed are means ± SE of GFP fluorescence normalized to OD$_{600}$ and background fluorescence from empty vector strains; n = 3. Asterisks denote significant difference based on two-sample *t*-test

comparison with DC3000 treated with fructose and aspartate, $P < 0.05$. Data are representative of three independent experiments.
(TIF)

**S2 Fig. Requirement of *aauS* and *aauR* for *hrpL* expression in DC3000 in response to type III secretion-inducing metabolites.** GFP fluorescence of DC3000 and Δ*aauS*Δ*aauR* carrying *hrpL*$_{promoter}$:*gfp* reporter plasmids. Bacteria were cultured for eight hours in minimal medium containing 10 mM fructose and with or without 200 µM glutamate, citrate, or 4-hydroxybenzoic acid. Graphed are means ± SD of the change in GFP fluorescence at T = 8 hours post-treatment. Fluorescence values were normalized to OD$_{600}$ and background fluorescence from empty vector strains; n = 12. Data are pooled from 3 independent experiments, n = 4 per experiment. Small-case letters denote statistical significance based on ANOVA with Tukey's post-hoc HSD test, $P < 0.05$.
(TIF)

**S3 Fig. Measurements of leaf bacteria populations at early timepoints used for assessing T3SS deployment and at later timepoints used for assessing bacterial growth. (A)** 1 x 10$^8$ cfu/mL of DC3000, Δ*aauR*, Δ*aauS* or Δ*aatJ* were syringe-infiltrated in Arabidopsis leaves. Leaf bacteria populations were measured by serial dilution plating of leaf extracts six hours post-infection. Graphed are means ± SE of bacteria colonies isolated from infected tissue, n = 3. Abbreviation ns is not significant based on ANOVA with Tukey's HSD, α = .05. **(B)** 5 x 10$^8$ cfu/mL of DC3000 or Δ*aauR* were syringe-infiltrated into Arabidopsis leaves. Leaf bacteria populations were measured by serial dilution plating of leaf extracts 6 hours post-infection. Graphed are means ± SE of bacteria colonies isolated from infected tissue, n = 3. Abbreviation ns is not significant based on *t*-test, $P$ = .444. **(C-D)** 1 x 10$^6$ cfu/mL of DC3000, Δ*aauS*, Δ*aauR*, or Δ*aatJ* and respective complemented strains were syringe-infiltrated into Arabidopsis leaves. Leaf bacteria populations were enumerated on day 0 and day 3 by serial dilution plating of leaf extracts. Graphed are log-transformed means ± SE of bacteria colony-forming units (cfus) isolated from infected tissue, n = 6 for day 0 and n = 8 for day 3. Small-case letters denote significance groupings based on ANOVA with Tukey's HSD, $P < 0.01$. Data shown were pooled from two independent experiments.
(TIF)

**S4 Fig. Loss of *aatJ*, *aauS* or *aauR* does not alter DC3000 growth in nutrient rich media.** DC3000, Δ*aatJ*, Δ*aauS*, and Δ*aauR* were individually inoculated into 100 µL of **(A)** Lysogeny broth (LB) or **(B)** King's B (KB) broth and grown for 24 hours at 28°C. OD$_{600}$ readings of each well were taken at 30 min intervals. Graphed are means ± SD of OD$_{600}$ readings, n = 3. Results are representative of 3 independent experiments.
(TIF)

**S5 Fig. Wild-type alleles of *aauS*, *aauR* or *aatJ* restore the growth of Δ*aauS*, Δ*aauR* and Δ*aatJ* strains in glutamate-supplemented medium. (A)** DC3000 and Δ*aauS* carrying empty vector (EV) pME6010 or *aauS*::pME6010; **(B)** DC3000 and Δ*aauR* carrying empty vector (EV) pME6010 or *aauR*::pME6010; and **(C)** DC3000 and Δ*aatJ* carrying empty vector (EV) pME6010 or *aatJ*::pME6010 were inoculated into 100 µL of M9 medium supplemented with 10 mM glutamate as the sole carbon source. Optical density of each culture was measured every 30 minutes from 6 to 18 hours post-inoculation. Graphed are means ± SD of OD$_{600}$ readings, n = 4. Results are representative of 3 independent experiments.
(TIF)

**S6 Fig. Loss of *aatP* does not alter DC3000 growth in rich media.** Timecourse analysis of DC3000 and Δ*aatP* growth in King's B (KB) broth. Cultures were inoculated at an optical density at λ = 600 nm ($OD_{600}$) of 0.05 and grown for 24 hours at 28°C. Graphed are means ± SD of $OD_{600}$ readings, n = 3. Results are representative of 3 independent experiments.
(TIF)

**S7 Fig. DC3000 *aatQ* and *aatM* genes are required for maximum growth on glutamate but are not required for glutamate-induced expression of T3SS genes or growth *in planta*.** Growth of DC3000 and Δ*aatQaatM* strains in KB medium **(A)** or M9 minimal medium with glutamate as a sole carbon and nitrogen source **(B)**. Cultures were inoculated at an optical density (λ = 600 nm) of 0.05 and grown for 24 hours at 28°C in wells of a 96-well plate. Optical density at λ = 600 nm ($OD_{600}$) readings of each well were taken at 30 min intervals. Graphed are means ± SD of $OD_{600}$ readings, n = 3. **(C)** Analysis of GFP fluorescence from DC3000 and Δ*aatQaatM avrRpm1*$_{promoter}$:*gfp* reporter strains incubated with 10 mM fructose (open bars) or 10 mM fructose and 200 μM glutamate (filled bars) for 6 hours. Graphed are means ± SD of GFP fluorescence normalized to $OD_{600}$ and background fluorescence from empty vector strains; n = 4. Abbreviation ns is not significant based on *t*-test with α = 0.05. **(D)** Growth of DC3000 and Δ*aatQaatM* strains in Arabidopsis. A 1 x $10^6$ cfu/mL inoculum of each strain was syringe-infiltrated into Arabidopsis leaves. Leaf bacteria populations were enumerated on day 0 and day 3 by serial dilution plating of leaf tissue extracts. Graphed are means ± SE of bacteria from 4 infected plants, n = 3 for day 0 and n = 4 for day 3. Abbreviation ns is not significant based on *t*-test with α = 0.05.
(TIF)

**S8 Fig. AauR directly interacts with Rbm-containing *aatJ* promoter DNA *in vitro*.** AlphaScreen assay of *E. coli*-expressed recombinant DC3000 AauR protein binding to 50 bp of *aatJ* promoter DNA including the Rbm sequence. Assays containing non-biotinylated DNA were included as negative controls to demonstrate specificity of detected interactions. Graphed are means ± SD of luminescence from assay wells, n = 3. Asterisks is *P* < 0.001 based on *t*-test. Data are representative of 3 independent experiments.
(TIF)

**S9 Fig. Multiple sequence alignment of the *hrpRS* promoter region containing Rbm from diverse *P. syringae* isolates.** Alignment of genomic DNA containing Rbm upstream of *hrpRS* from 38 *P. syringae* isolates. Sequences are labeled with phylogroup and strain name. Full names can be found in S3 Table. Upper and lower case letters below alignment are consensus sequence of nucleotides based on 100% and ≥85% identity, respectively; dots on the consensus line indicate no consensus based on either criteria. Highlighted in blue is AauR-binding motif (Rbm).
(TIF)

**S10 Fig. A maximum likelihood phylogenetic tree of *hrpR* sequences indicates inheritance through vertical descent.** Nucleotide sequences of *hrpR* from all canonical tripartite type III secretion system encoding *P. syringae* strains examined in this study are included in the phylogeny. Ultrafast bootstraps (>95% support) calculated by IQ-TREE are reported. Within phylogroup supports are all above the 95% bootstrap support threshold. Branch lengths indicate the number of mutations per site. The tree is artificially rooted on representatives from PG9 for reference.
(TIF)

**S11 Fig. A maximum likelihood phylogenetic tree of intergenic sequences between *hrpH* and *hrpR* indicates inheritance through vertical descent.** Nucleotide sequences corresponding to the intergenic regions between *hrpH* and *hrpR* from all canonical tripartite type III secretion system encoding *P. syringae* strains examined in this study are included in the phylogeny. Ultrafast bootstraps (>95% support) calculated by IQ-TREE are reported. Within phylogroup supports are all above the 95% bootstrap support threshold. Branch lengths indicate the number of mutations per site. The tree is artificially rooted on representatives from PG9 for reference.
(TIF)

**S12 Fig. A maximum likelihood phylogenetic tree of *aatJ* sequences indicates inheritance through vertical descent.** Nucleotide sequences of *aatJ* from diverse *P. syringae* isolates and other pseudomonads are included in the phylogeny. Ultrafast bootstraps (>95% support) calculated by IQ-TREE are reported. Within phylogroup supports are all above the 95% bootstrap support threshold. Branch lengths indicate the number of mutations per site. The tree is artificially rooted on branches with representative *Pseudomonas* spp. outside of *P. syringae* for reference.
(TIF)

**S13 Fig. A maximum likelihood phylogenetic tree of AauS sequences indicates inheritance through vertical descent.** Amino acid sequences of AauS from diverse *P. syringae* isolates and other pseudomonads are included in the phylogeny. Ultrafast bootstraps (>95% support) calculated by IQ-TREE are reported. Within phylogroup supports are all above the 95% bootstrap support threshold. Branch lengths indicate the number of mutations per site. The tree is artificially rooted on branches with representative *Pseudomonas* spp. outside of *P. syringae* for reference.
(TIF)

**S14 Fig. Loss of AauR does not alter B728a growth in rich medium LB or M9 minimal medium with glucose and ammonium chloride.** Timecourse analysis of B728a and Δ*aauR* growth in (**A**) standard M9 medium supplemented with glucose and ammonium chloride or (**B**) lysogeny broth (LB). Cultures were inoculated at an optical density at λ = 600 nm ($OD_{600}$) of 0.05 and grown for 24 hours at 28˚C in a Tecan Spark 10M plate reader. Graphed are means ± SD of $OD_{600}$ readings, n = 3. Results are representative of 3 independent experiments.
(TIF)

**S15 Fig. Hypothetical model for role of AatJ-AauS-AauR in activating type III secretion in response to plant-derived metabolite signals.** Plant-exuded aspartate and glutamate enter the periplasm, mostly likely through an outer membrane porin, and bind to AatJ, an ABC transporter-associated substrate-binding protein. AatJ, in complex with aspartate or glutamate, binds to AatQM to deliver these metabolites for transport across the inner membrane, and also binds to AauS to activate signaling of the two-component system AauSR. Alternatively, aspartate/glutamate may directly bind to and activate AauS, with AatJ functioning indirectly in promoting AauSR activation (not shown in model). Activated AauS phosphorylates and activates AauR, which in turn binds to a conserved motif (AauR binding motif, Rbm) present within the promoter region of *aatJ* to increase expression of *aat*/*aau* genes. Activated AauR also binds to an Rbm upstream of the operon encoding enhancer-binding proteins HrpR and HrpS. Increased abundance of HrpR and HrpS results in increased expression of the gene encoding the alternative sigma factor HrpL, which in turn activates expression of genes encoding type III secretion system (T3SS) structural components and effectors. See discussion for

additional details of model.
(TIF)

**S1 Table. Bacterial strains and plasmids used in this study.**
(PDF)

**S2 Table. Sequences of oligonucleotides used in this study.**
(PDF)

**S3 Table. Metadata of *P. syringae* genomes and individual gene sequences used for phylogenetics.**
(PDF)

**S1 Data. Excel spreadsheet containing the underlying numerical data for figure panels.**
(XLSX)

## Acknowledgments

We thank Tom Wolpert and Jeff Chang for helpful discussions and critical reading of this manuscript, and the Department of Botany and Plant Pathology at Oregon State University for its generous support of the computing cluster.

## Author Contributions

**Conceptualization:** Qing Yan, Jeffrey C. Anderson.

**Data curation:** Qing Yan, Conner J. Rogan, Yin-Yuin Pang, Edward W. Davis, II, Jeffrey C. Anderson.

**Formal analysis:** Qing Yan, Yin-Yuin Pang, Edward W. Davis, II, Jeffrey C. Anderson.

**Funding acquisition:** Jeffrey C. Anderson.

**Investigation:** Qing Yan, Conner J. Rogan, Yin-Yuin Pang, Edward W. Davis, II, Jeffrey C. Anderson.

**Project administration:** Jeffrey C. Anderson.

**Supervision:** Jeffrey C. Anderson.

**Writing – original draft:** Qing Yan, Edward W. Davis, II, Jeffrey C. Anderson.

**Writing – review & editing:** Qing Yan, Conner J. Rogan, Yin-Yuin Pang, Edward W. Davis, II, Jeffrey C. Anderson.

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
