## [Decision Letter · Decision Letter 0]

23 Apr 2020

Dear Dr Anderson,

Thank you very much for submitting your manuscript "Ancient co-option of an amino acid ABC transporter locus in Pseudomonas syringae for host signal-dependent virulence gene regulation" for consideration at PLOS Pathogens. As with all papers reviewed by the journal, your manuscript was reviewed by members of the editorial board and by several independent reviewers. The reviewers appreciated the attention to an important topic. Based on the reviews, we are likely to accept this manuscript for publication, providing that you modify the manuscript according to the review recommendations.

As you prepare a revised version of the manuscript, please consider the following:

1. More detail describing the screen would be useful, including indication of other genes identified.

2. The key gene induction assays in figure 1 would be strengthened by addition of a genetic negative control (e.g. a hrpR/S mutant) in addition to the sucrose only treatment.

3. Inclusion of a genetic negative control also would provide better context for the magnitude of growth defects in the key bacterial growth assay in figure 2C.

4. Rather than representative, single experiments, data for bacterial growth should be a composite of multiple, independent experiments.

5. Phospho-point mutants requested by reviewer #2 are beyond the scope expected for this study.

6. Presenting in vitro bacterial growth as the OD600 without log transformation is OK.

Sincerely,

David Mackey

Associate Editor

PLOS Pathogens

Bart Thomma

Section Editor

PLOS Pathogens

Kasturi Haldar

Editor-in-Chief

PLOS Pathogens

orcid.org/0000-0001-5065-158X

Michael Malim

Editor-in-Chief

PLOS Pathogens

orcid.org/0000-0002-7699-2064

Reviewer Comments (if any, and for reference):

Reviewer's Responses to Questions

**Part I - Summary**

Reviewer #1: The manuscript by Yan et al. identifies a two-component system and substrate binding protein that contributes to the regulation of the type III secretion system in Pseudomonas syringae. The system is conserved across P. syringae species and regulates the type III secretion system in response to host metabolites (amino acids aspartate and glutamate). Overall the manuscript is thorough and well written. I only have minor comments.

Reviewer #2: The authors have identified a two component system and associated SBP that drive full expression of hrp regulon and P syringae virulence in response to acidic amino acids through direct regulation of hrpR transcription. The aauS sensor kinase, aauR response regulator, and aatJ SBP all play similar non-additive roles in contributing to hrp regulon expression in response to asp and glu but hrp regulation does not require the ABC transporter genes for asp/glu in vitro or in vivo. The authors find a consensus aauR box (Rbm) upstream of hrpR and demonstrate that aauR binding to the hrpR promoter is dependent on the Rbm. Rbm mutants phenocopy an aauR mutant indicating that the aauR hrp regulation phenotypes can be adequately explained by the hrpR Rbm site. The hrpR Rbm site is (almost) perfectly conserved among P. syringae in all phylogroups with a canonical T3SS. Similar mutant phenotypes were observed with both Pto and Pss. The appropriateness of glu/asp as an accurate signal of the plant host environment and the potential evolutionary context of AauR/S-HrpR/S regulation is discussed.

Reviewer #3: This study reveals a novel regulatory mechanism controlling the expression of the Type III secretion system in the bacterial pathogen Pseudomonas syringae. The study is thorough and complete and makes a significant new contribution to our understanding of the complex mechanisms by which plant pathogens sense and respond to their environment to regulate expression of virulence factors. The amount of data presented is substantial, the experiments are carefully done and well-controlled, and the manuscript is clearly written. Overall, the claims are well-supported by the data, and as described below, I have only a few scientific and editorial comments that the authors should consider when revising their manuscript.

**Part II – Major Issues: Key Experiments Required for Acceptance**

Reviewer #1: (No Response)

Reviewer #2: This is one of those excellent papers where, even with a critical eye, I find very little to fault with this work. The genetics are solid with appropriate complementation and controls. The experiments are carefully and thoroughly designed. Multiple forms of evidence are used to support major points both in vivo and in vitro.

To complement the Rbm analysis I would like to see an analysis of genetic variation of the aatJ-aauR locus within the same P. syringae as well as in other pseudomonads.

Other that that, I have only minor concerns.

Reviewer #3: 1. Fig. 2A. How is expression of AvrPto in planta normalized? Were bacterial numbers in the leaf tissue analyzed from the same samples?

**Part III – Minor Issues: Editorial and Data Presentation Modifications**

Reviewer #1: Regarding the screen

-was this screen successful in finding components that are already known to play a role in general T3SS/T3E expression?

-how was reduced GFP fluorescence scored/what was the cutoff?

-would intermediate levels of GFP fluorescence correspond to mutations that play intermediate roles in T3SS expression?

Figures

Figure 2C- how many leaves per plant for growth assays?

Figure 2D- based on the symptoms in the one picture shown, we might expect a larger difference in bacterial growth (completely green vs. chlorotic). Are these representative?

Same for Figure 5F and G.

Figure 3- How close are the amino acid concentrations used in this study to physiological concentrations (aspartate, glutamate, serine, glutamine)?

Figure S5; lines 191-193, “Growth of all three mutant strains could be restored by introduction of plasmids carrying respective wild type alleles of each mutated gene (S5 Fig).” aauR complementation doesn't seem to show this exactly, growth rate starts the same, but then diverges from DC3000? Explain.

Overall it would have been ideal to have mutant in a known type III secretion system regulator as a control for comparison since the aauS, aauR and aatJ mutants seem intermediate in phenotype (eg. HrpL mutant)

Reviewer #2: Why were only 14 of the 20 L amino acids tested? Not a huge deal but it's an odd omission. Likewise a good positive control could have been added to AlphaScreen protein-DNA interaction experiment. Engineered phospho-residue point mutations in aauS and aauR would have lent strong support for the model. Lastly, I would have preferred to see more explicit descriptions of hypotheses for the role of AatJ in the regulatory cascade.

references 17 and 51 are duplicates

Reviewer #3: Minor Scientific comments

2. Figure 1. It is interesting that the reduction of expression of hrpR/Sprom-gfp in the aau mutants is less dramatic that for expression of avrRpm1 and hrpL. Do you have any idea as to why this is the case?

3. Line 168 and Fig S4.The conclusion that the decreased growth of the mutants in in leaf tissue is not due to a general fitness defect cannot be made based on the observation that no difference in growth of the mutants was detected in KB or LB. Growth of the mutants should also be carefully assessed in defined media that may more closely resembles conditions in the apoplastic space. I think the relevant growth information is presented in a later figure (fig 3D?). Perhaps that data could be presented or referred to earlier in the results section?

4. Fig 3c and others. Bacterial growth, which is exponential, should be plotted on a logarithmic scale. All of the in-culture growth curves are plotted on a linear scale. I don't believe this is appropriate.

5. Fig. 3C. The growth delay observed for the aauR, aauS and aatJ mutants in M9 + glutamate is interesting, and suggests that the aat/aau locus is required in part for the normal uptake of extracellular amino acids. Why is there only a delay in growth, as opposed to a more severe inhibition of groth? Is an aat/aau-independent mechanism induced to compensate for the absence of aat/aau?

6. Figs 5A and 6A: the AauR binding motif is located quite a bit upstream of the translation start site for hrpR. This seems like a very long distance away for a regulatory motif. Does this suggest that there is a long 5’UTR? Or a binding site at a distance from the promoter, with possible DNA looping? Is there any data in the public domain that might provide you information about the transcription start site for the hrpR/S operon (RNA seq read, perhaps)? This is not a major concern, but some discsussion about the long distance should be included, especially since it is strongly conserved amongst the P. syringae genomes analyzed here.

7. The hypothesis that AauS-AauR regulation of hrpRS may predate P. syringae speciation is intriguing. Could it also predate the speciation of P. syringae from other plant pathogenic Pseudomonas species?

Editorial comments

1. The manuscript is well-written, engaging and easy to read. However, I did not find that enough basic information regarding how the mutant screen was carried out is provided in the Results section. I had to piece this together from the methods, and from referring to Turner et al (Ref. 18). I realize that PLoS journals aim for shorter manuscripts, but I wish more info had been provided up front.

2. I suggest using the terms Microbial associated molecular patterns (MAMPs) rather than PAMPs.

3. Fig 2 figure legend. I suggest adding the qualifier “normal level of virulence” between “….required for” and “DC3000 virulence”. A similar change should be made for the legend for Fig. 5. The mutants exhibit only a partial reduction in virulence.

4. Figure 5E. This graph is hard to interpret, as one cannot see any data point value other than the biotin-DNA in the hrpRS strain. Can the axis be altered to show the low values? Or “nd” for none-detected if that is the case? It is hard to know how many data points are actually presented in the graph.

5. Line 390. Is OD600 = 5.0 correct? Also, why was the culture shaken for four weeks? This seems like a very long time to grow a bacterial culture.

6. Inclusion of a diagram to illustrate the model for how AauS/AauR-dependent mechanism regulates expression of T3SS, to accompany the las section of the discussion, would be a nice addition to the manuscript.

PLOS authors have the option to publish the peer review history of their article (what does this mean?). If published, this will include your full peer review and any attached files.

Reviewer #1: No

Reviewer #2: No

Reviewer #3: Yes: Barbara Kunkel
---

## [Editor Report · Decision Letter 1]

3 Jun 2020

Dear Dr Anderson,

We are pleased to inform you that your manuscript 'Ancient co-option of an amino acid ABC transporter locus in Pseudomonas syringae for host signal-dependent virulence gene regulation' has been provisionally accepted for publication in PLOS Pathogens. Congrats Jeff, et al.!

Best regards,

David Mackey

Associate Editor

PLOS Pathogens

Bart Thomma

Section Editor

PLOS Pathogens

Kasturi Haldar

Editor-in-Chief

PLOS Pathogens

orcid.org/0000-0001-5065-158X

Michael Malim

Editor-in-Chief

PLOS Pathogens

orcid.org/0000-0002-7699-2064
---

## [Editor Report · Acceptance letter]

6 Jul 2020

Dear Dr Anderson,

We are delighted to inform you that your manuscript, "Ancient co-option of an amino acid ABC transporter locus in *Pseudomonas syringae* for host signal-dependent virulence gene regulation," has been formally accepted for publication in PLOS Pathogens.

Best regards,

Kasturi Haldar

Editor-in-Chief

PLOS Pathogens

orcid.org/0000-0001-5065-158X

Michael Malim

Editor-in-Chief

PLOS Pathogens

orcid.org/0000-0002-7699-2064